# Melatonin and Carbohydrate Metabolism in Plant Cells

**DOI:** 10.3390/plants10091917

**Published:** 2021-09-15

**Authors:** Marino B. Arnao, Josefa Hernández-Ruiz, Antonio Cano, Russel J. Reiter

**Affiliations:** 1Department of Plant Biology (Plant Physiology), Faculty of Biology, University of Murcia, 30100 Murcia, Spain; jhruiz@um.es (J.H.-R.); aclario@um.es (A.C.); 2Department of Cell Systems and Anatomy, UT Health, Long School of Medicine, San Antonio, TX 78229, USA

**Keywords:** carbohydrates, melatonin, phytomelatonin, primary metabolism, starch, sucrose

## Abstract

Melatonin, a multifunctional molecule that is present in all living organisms studied, is synthesized in plant cells in several intercellular organelles including in the chloroplasts and in mitochondria. In plants, melatonin has a relevant role as a modulatory agent which improves their tolerance response to biotic and abiotic stress. The role of melatonin in stress conditions on the primary metabolism of plant carbohydrates is reviewed in the present work. Thus, the modulatory actions of melatonin on the various biosynthetic and degradation pathways involving simple carbohydrates (mono- and disaccharides), polymers (starch), and derivatives (polyalcohols) in plants are evaluated. The possible applications of the use of melatonin in crop improvement and postharvest products are examined.

## 1. Introduction

Plants obtain their energy and resources via an autotrophic means. All their organic molecules are synthesized from inorganic elements such as CO_2_ and primarily from diverse compounds including nitrogen, sulfur, and phosphorus, among others. In addition to their rich secondary metabolism, plants produce a multitude of primary metabolites including carbohydrates, lipids, and amino acids. The group of carbohydrates of plant origin comprises a wide range of simple sugars such as mono- and disaccharides, sugar alcohols, and polymers such as starch and cellulose [1,2].

Pathways of plant carbohydrate metabolism are well known. From the generated-Calvin cycle triose-phosphate pool, the biosynthetic routes of fructose, glucose, and other simple carbohydrates are well delineated. In plants, sucrose metabolism is crucial to feed phloem transport from source parts to sink parts. Gluconeogenesis and pentose phosphate shunt are the major pathways where simple carbohydrate biosynthesis is involved [3]. Other important compounds such as polymers (starch, cellulose, and derived) and sugar alcohols/polyalcohols (glycerol, myo-inositol, sorbitol, manitol, etc.) are synthesized from the formers. Carbohydrates, polyalcohols, and some amino acids (especially proline), in addition to their nutritional function in cells, have an interesting role as osmoregulatory compounds, especially in stressful situations [4,5].

The present paper highlights a literature summary of the effects of melatonin on carbohydrates metabolism, focusing on diverse aspects such as carbohydrate content, gene-related regulation, and the possible use of melatonin to improve crop production and quality and postharvest preservation.

## 2. Biosynthesis of Melatonin in Plants

Melatonin (*N*-acetyl-5-methoxytryptamine) is a tryptophan-derived compound discovered in plants in 1995 [6,7,8]. Melatonin is a highly studied biomolecule due to its known role in mammals as a regulating hormone of sleep-wake cycles, and other functions in endogenous rhythms, mood, metabolism, and immunological responses [9,10]. In addition, it has been investigated as to its therapeutic efficacy in Alzheimer’s disease, Parkinsonism, cancer, diabetes, and SARS-CoV-2 [11,12,13,14,15,16,17].

Melatonin biosynthesis in plants originates with the amino acid tryptophan, which is endogenously synthesized in plant cells in the chorismate pathway. Five enzymes are involved in the conversion of tryptophan to melatonin; these are tryptophan decarboxylase (TDC), tryptamine 5-hydroxylase (T5H), serotonin *N*-acetyltransferase (SNAT), acetylserotonin methyltransferase (ASMT), and caffeic-*O*-methyltransferase (COMT) [18,19]. These enzymes catalyze the conversion of the indolic compounds tryptophan, tryptamine, serotonin, 5-methoxytryptamine, and *N*-acetylserotonin to melatonin, as illustrated in the biosynthetic pathway shown in Figure 1. However, this primary melatonin biosynthetic pathway may present alternatives such as serotonin biosynthesis through 5-hydroxytryptophan, although this possibility seems specific to animals since the responsible enzyme (tryptophan hydroxylase) has not been detected in plants. In addition, a conversion of *N*-acetylserotonin to serotonin by the enzyme *N*-acetylserotonin deacetylase has been described [18,20]. With respect to the subcellular localization, several studies in arabidopsis and rice plants indicated that the involved enzymes act in the cytoplasm (TDC, ASMT and COMT), endoplasmic reticulum (T5H), and chloroplasts (SNAT) [21]. In addition, the participation of mitochondria has been described, through arylalkylamine *N*-acetyltransferases (AANAT) and hydroxyindole-*O*-methyltransferases (HIOMT), observing that, when the melatonin pathway is artificially blocked in chloroplasts, melatonin biosynthesis shifts to the mitochondria to maintain melatonin generation [21,22]. Generally, stressors induce melatonin biosynthesis in plants through the upregulation of diverse biosynthesis isozyme transcripts, increasing endogenous melatonin production [23].

## 3. Roles of Melatonin in Plants

Melatonin is a pleiotropic molecule in plants. Melatonin has many beneficial actions, generally improving physiological responses such as seed germination and growth, photosynthesis (pigment content, photorespiration, stomatal conductance and water economy), seed and fruit yield, osmoregulation, and the regulation of the different metabolic pathways (carbohydrates, lipids, nitrogen compounds, sulphur, and phosphorus cycles) [24,25,26,27,28,29,30,31,32,33,34]. With respect to secondary metabolism, melatonin induces the biosynthesis of simple phenols, flavonoids, anthocyanins, carotenoids, and several terpenoids [35,36,37,38]. Melatonin promotes rooting processes [39,40,41,42,43] and also delays leaf senescence [44,45,46,47,48,49]. In postharvest fruit, it regulates ethylene and lycopene content, as well as general ripening metabolism and induces parthenocarpy during fruiting [50,51,52]. It also preserves cut flowers [53,54]. In pathogen infections, melatonin slows damage, stimulating systemic acquired resistance (SAR) and contributes to crop health [55]. Due to this high number of actions, melatonin has been referred to as a plant master regulator [56,57], mainly due to its role as a plant hormone regulator, with a substantial influence on auxin, gibberellins, cytokinins, abscisic acid, ethylene, jasmonic acid, salicylic acid, and brassinosteroids [58,59].

Melatonin displays a relevant role in the stress responses. Similar to what occurs in animal cells, melatonin acts as an excellent scavenger of reactive oxygen species (ROS) and reactive nitrogen species (RNS) in plants. This antioxidant capacity has been extensively studied [60,61,62]. The data show that melatonin acts as a direct antioxidant, neutralizing several ROS/RNS and other radical species harmful to the cell, and also acts as an activator of the antioxidant response, upregulating various transcription factors that trigger the activity of antioxidant enzymes such as superoxide dismutases, catalases, peroxidases, and those involved in the ascorbate-glutathione cycle, among others [22,63]. Via these means, melatonin acts as a master regulator of the responses of the redox, hormonal, and osmoregulatory systems [56,58,59,64]. In summary, as can be seen in Figure 2, through the redox and hormonal network, melatonin regulates photosynthesis, primary and secondary metabolism, and pathogenic response to increase abiotic/biotic tolerance and, as a result, crop yield. One of the most interesting aspects is the ability of melatonin to regulate the carbohydrate metabolism and its relationship with the osmoregulatory response, which is a key in stressful situations of plants.

## 4. Effect of Melatonin in Simple Carbohydrates, Starch, and Polyalcohols

The term phytomelatonin refers to melatonin of plant origin as opposed to the animal hormone, but they have the identical chemical structure. The first studies on the role of phytomelatonin in plants appeared at the end of the last century and the beginning of the present one [65]. Table 1 summarizes the results of studies on melatonin and carbohydrates in plants. Based on these data, the initial report related to melatonin and carbohydrates in plants is an in vitro study in cherry rootstock. In this study, exogenous melatonin added to the culture media induced plant growth and rooting in shoot tip explants; in addition, an elevation in endogenous levels of total soluble sugars in 9-week-old plants, both in leaves and roots, and in chlorophylls, carotenoids, and proline level were also observed. These findings indicate an improvement in plant primary carbon metabolism, with a melatonin-concentration dependent response [31]. Also in apple trees, melatonin treatment of leaves produced an increase in the levels of monosaccharides, sucrose, starch, and sorbitol as well as an improvement in the photosynthetic rate and a reduction in foliar senescence and autophagy [32]. Other studies were focused on improving the plants’ tolerance to certain stresses. Thus, melatonin treatments enhanced saline tolerance in soybean [33], tomato [66], and bermudagrass plants [67] (see Table 1), accompanied by an activation of carbohydrate metabolism and, in some cases, lipid and ASC-GSH metabolism as well [68]. There are many studies on the promotional effect of fruit development after the application of melatonin in leaves and/or roots. One of the first was carried out in tomato plants, where melatonin applications induced photosynthetic processes with a higher yield in biomass and a greater number of fruits which were of greater caliber and exhibited optimal ripening [69]. In pear trees, 100 µM melatonin treatments induced higher total sugars and starch levels and better fruit sizes which were of high quality [70]. In addition, postharvest melatonin treatments in various fruits gave rise to higher quality fruits with an increased content of sugars, starch, organic acids, and pigments, as had been demonstrated in tomato [71] and banana [72], and other fruits such as peach, strawberry, pear, plum, and litchi [27,53]. In one comprehensive study, melatonin treatments induce innate immunity in *Arabidopsis* with the accumulation of various sugars and glycerol, as well as increasing disease resistance against *Pseudomonas syringe* [73]. In general, plants treated with melatonin exhibit increases in the levels of simple sugars, sucrose, starch, and some polyalcohols.

## 5. Regulatory Action of Melatonin on Carbohydrate Metabolism

In general, melatonin improves photosynthetic and related parameters, such as photosynthetic rate, transpiration rate, stomatal conductance, leaf area, relative water content, and levels of chlorophylls and carotenoids, and also delays leaf senescence. Melatonin has a protective role against oxidative stress, reducing the levels of superoxide anion, hydrogen peroxide, and malondialdehyde, and improving membrane stability indexes. It also induces the expression of genes for antioxidant enzymes such as superoxide dismutases, catalases, guiacol-, and ascorbate peroxidases, which in turn raises ascorbate and glutathione levels. Also relevant is the melatonin-mediated improvement in the uptake of mineral nutrients, which induces the expression of mineral transporters. Collectively, the up- and downregulated genes following melatonin treatment functions to aid plants in physiologically overcoming negative stress situations and to increase tolerance to multiple abiotic stressors such as drought, waterlogging, salinity, heavy metals, extreme temperatures, radiation, etc., including osmoregulatory responses [24,26,46,56,64,84,85,86,87,88,89].

In 2014, Guo and colleagues performed transcriptional studies and were the first to detect changes in the expression of genes involved in carbohydrate metabolism due to melatonin treatment [90,91]. In addition, in an excellent study on the effect of melatonin in salinity tolerance of soybean plants, a detailed transcriptomic analysis on primary metabolism was presented. Melatonin clearly over-expressed the transcripts of many enzymes related to photosynthesis, starch, sucrose, glycolysis, fermentation, the Krebs cycle, and other metabolic pathways [33]. Figure 3 diagrammatically summarizes some of the genes up or downregulated by melatonin that are related to carbohydrate metabolism.

In bermudagrass, melatonin-pre-treated plants exhibited significantly higher levels of several metabolites than non-treated plants under abiotic stress conditions (salinity, drought, and cold). These primary metabolites included 10 amino acids, 18 carbohydrates (allose, arabinose, cellobiose, fructose, galactose, gentiobiose, glucose, lactose, lactulose, maltose, manobiose, manose, melibose, sucrose, tagatofuranose, talose, trehalose, and turanose) and five polyalcohols (dulcitol, galactinol, glycerol, myo-inositol, and sorbitol), involved in glycolysis, oxidative pentose phosphate pathway, and the Krebs cycle [67,75]. With respect to gene regulation, 2361 upregulated and 1572 downregulated transcripts were differentially expressed in melatonin-treated plants *versus* control. Some examples of carbohydrate-related transcripts regulated by melatonin are: glycosyl –transferases, -hydrolases, -phosphatases, -epimerases, -invertases and -mutases, hexokinases, mannosidases, α- and β-amylases, α- and β- glucan related-enzymes, and several dehydrogenases (3-phosphoglycerate-, UDP-glucose-, alcohol- and aldehyde-), among others.

The regulation of carbohydrate metabolism by melatonin is accompanied by an activation of chloroplast metabolism and an improvement in the function of these organelles. As seen in Figure 3, there are many factors upregulated by melatonin in the photosynthetic light apparatus and in the Calvin cycle [33,67,75]. Melatonin also stimulates the biosynthesis and mobilization of starch and of sucrose destined for the phloem. The role of melatonin in sucrose metabolism has received significant attention. In maize plants, low doses of melatonin (1–10 μM) induce sugar metabolism, photosynthesis, and sucrose phloem loading. The authors demonstrated that high doses of melatonin inhibit seedling growth by inducing the excessive accumulation of sucrose, hexose and starch, suppressing photosynthesis and sucrose phloem loading [76]. The role of melatonin in improving sucrose biosynthesis was also confirmed in pear, grape, and rice plants [70,79,80].

Intimately related to carbohydrates is the osmoregulatory response in plants subjected to stress. Melatonin clearly mediates these responses by increasing the levels of carbohydrates and polyalcohols, as already described. One of the key metabolites in the osmoregulatory response is proline, an amino acid that accumulates in the presence of various stressors, especially drought and salinity [92,93]. Melatonin raises proline levels in stressed plants (Figure 3), which has been demonstrated in various species and situations [83,94,95]. Several reviews in this regard can be consulted since this aspect is outside the present review on carbohydrates [26,64,84,85,96,97,98].

## 6. Conclusions

Numerous investigations have provided data on the regulatory role of melatonin in multiple metabolic pathways in plants. In primary metabolism, its critical action on enzyme transcripts and regulatory factors in different organelles (chloroplasts, mitochondria, endoplasmic reticulum) and subcellular sites (cytosol, cell wall) stands out. Metabolic processes such as photosynthesis, the pentose phosphate shunt, gluconeogenesis, glycolysis as well as the Krebs cycle and the biosynthesis of amino acids and fatty acids are clearly under the influence of melatonin at several key steps. Carbohydrate metabolism is one of the most studied, although much remains to be known. From the regulation of Rubisco to the processes of glycolysis and fermentation, melatonin appears to play a decisive role in the fate of carbohydrates synthesized in the chloroplast and cytosol. Thus, melatonin regulates the production of triose phosphate in the Calvin cycle, its transformation into hexoses and also the pool of starch in the chloroplastic stroma and that of sucrose in the cytosol and cell walls. In general, melatonin activates the primary metabolism, both of carbohydrates and of other primary components such as lipids and amino acids. The result is an activation of the metabolic turnover such that it is adequate and conditioned to the physiological situation of the moment. In summary, melatonin has multiple regulation actions; for example, it influences photosynthesis, improving the efficiency of Rubisco and other Calvin cycle-related enzymes, Photosystem I and II, chlorophyll and carotenoid content and stomatal complex, with the result of a higher net photosynthesis, and, in specific carbohydrate metabolism, melatonin mobilizes some key pathways such as starch and sucrose biosynthesis, through SPS, SuS, and invertases upregulation, mainly (see Figure 3), increasing biosynthesis sugar rate to cope with stressful situations [99].

In parallel, melatonin regulates many factors of the metabolism of plant hormones that, together with the modulation of the redox network, make melatonin an essential biostimulator or plant growth regulator, leading the plant through its functions to an adaptation to environmental situations against adverse effects and increasing tolerance to stressors [58].

Regarding melatonin’s possible applications in crop improvement and postharvest actions, there are already many published examples (Table 1) [27,100,101]. The ability of melatonin treatment to modify carbohydrate metabolism and increase the levels of sugars in fruits and their organoleptic qualities are a result of its capacity to influence many stages of secondary metabolism, especially in phenolic compounds and terpenes biosynthesis. Highlights include its regulatory role on anthocyanins and other flavonoids, as well as carotenoids and essential oils [35,36,37,38]. Additionally, melatonin treatment positively affects crop yield; an increased production as a result of melatonin treatment has been observed for rice, wheat, cucumber, tomato, rapeseed, and others [27,102,103,104]. Obviously, there are many aspects to be investigated relative to the influence of melatonin on carbohydrate metabolism, such as: the regulatory action of genes in the nucleus, chloroplasts and mitochondria; its interactions with other plant hormones; its functions in different organs (leaf, stem, root, flowers, fruits); its action on the accumulation and degradation of starch in amyloplasts; its ability to influence the metabolism of sucrose in source and sink tissues, thereby regulating phloem loading and unloading; and its action in the regulation of the biosynthesis of polyalcohols and proline, which is key to understanding the osmoregulatory response to stress. Finally, a complete understanding of its role in carbohydrates/fatty acid/amino acid balance has yet to be realized.

## Figures and Tables

**Figure 1 plants-10-01917-f001:**
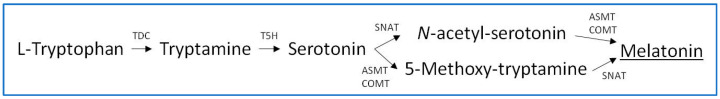
Biosynthesis of melatonin in plants.

**Figure 2 plants-10-01917-f002:**
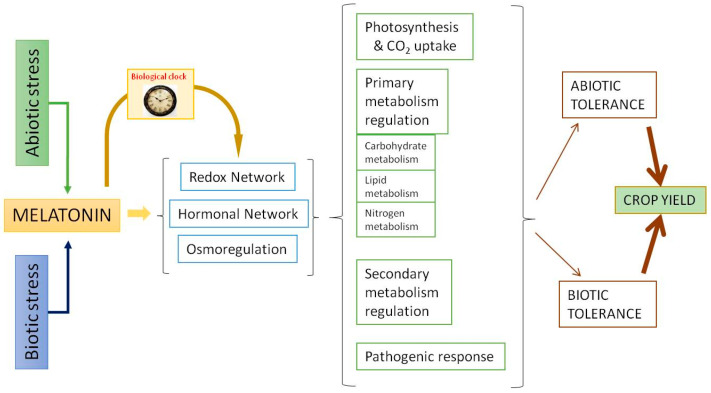
Melatonin actions as a response to abiotic and biotic stressors.

**Figure 3 plants-10-01917-f003:**
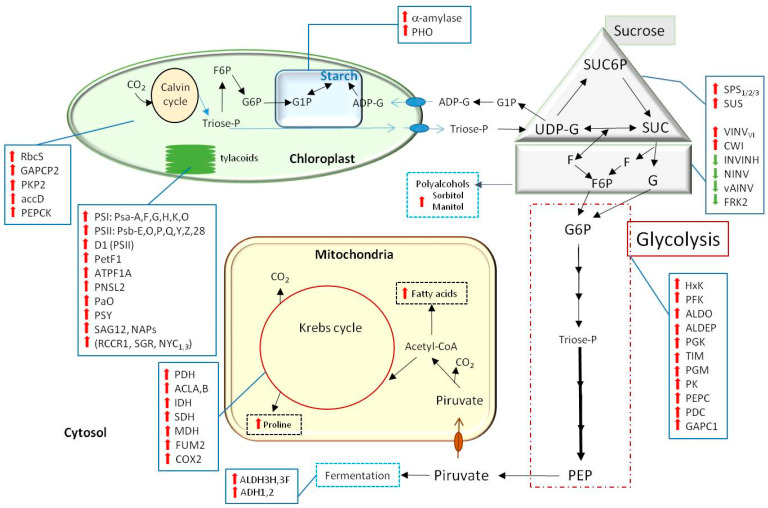
Schematic model representing melatonin’s action in different metabolic pathways. Melatonin regulates several enzymes and transcripts related with carbohydrates at different subcellular levels. Arrows indicate: 
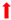
, increased level of transcript expression and 
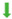
, decreased level, or transcript expression (see abbreviations in text).

**Table 1 plants-10-01917-t001:** Examples of studies on carbohydrates and melatonin.

Plant	Melatonin Treatment (µM)	Compound Level vs. Un-Treated	Response vs. Un-Treated	Reference
*Prunus avium x* *Prunus cerasus* (in vitro)	0.05–10	↑ total carbohydrates	↑ rooting ↑ plant biomass	[31]
*Malus hupehensis* tree	100	↑ fructose, glucose, sucrose, starch ↑ sorbitol	↑ photosynthesis ↓ senescence ↓ autophagy	[32]
Tomato fruits	1–500	↑ soluble sugars	↑ fruit ripening and quality	[71]
Tomato plants	100	↑ glucose, sucrose, inositol ↓ fructose, galactose	↑ photosynthesis ↑ plant biomass ↑ fruit number and size	[69]
	20–50	↑ soluble sugars ↑ ascorbate and GSH	↑ photosynthesis ↑ plant growth ↑ NaCl tolerance	[66] [74]
Soybean	50 and 100	↑ carbohydrate metabolism, fatty acid biosynthesis, and ascorbate metabolism ↑ light reactions, Calvin cycle, carbohydrate, amino acid, fatty acid metabolism and Krebs cycle	↑ germination, biomass ↑ photosynthesis ↑ cell division ↑ NaCl tolerance	[33]
Bermudagrass *(Cynodon dactylon)*	4–100	54 metabolites, including amino acids, organic acids, sugars, and sugar alcohols ↑ photosyntesis, Calvin cycle and carbohydrate metabolism	↑ NaCl tolerance ↑ cold tolerance ↑ drought tolerance	[67]
	100	↑ arabinose, mannose, gluco-pyranose, maltose and turanose	↑ cold tolerance ↑ photosynthesis	[75]
Maize	10–100	↑ fructose, glucose, sucrose, starch and its biosynthesis genes	↑ photosynthesis ↑ leaf and root growth	[76]
	10–1000	↑ total soluble sugars ↑ nitrogen compounds ↑ expressions of genes involved in C- and N- metabolisms	↑ photosynthesis ↑ plant growth	[77]
Banana fruits	50–500	↑ total soluble sugars ↑ starch	↑ fruit ripening and quality ↓ ethylene	[72]
*Vicia faba*	50	↑ soluble sugars ↑ ascorbate and GSH	↑ As tolerance ↑ photosynthesis ↑ plant growth	[68]
*Brassica juncea*	10–50	↑ total soluble sugars ↑ reducing sugars	↑ photosynthesis ↑ plant growth ↑ mineral nutrition	[78]
Grape plants	50–200	↑ fructose, sucrose, starch, reducing sugars ↑ sucrose biosynthesis genes	↑ photosynthesis ↑ plant growth ↑ mineral nutrition	[79]
Rice plants	20	↑ fructose, sucrose, starch, reducing sugars ↑ sucrose biosynthesis genes	↑ As tolerance ↑ Krebs cycle	[80]
Pear tree	100	↑ total soluble sugars ↑ sucrose, starch, reducing sugars, sorbitol ↑ sucrose synthase, invertases	↑ photosynthesis ↑ fruit size and quality	[70]
*Malus domestica* (plants)	1000	↑ fructose, glucose, sucrose, sorbitol ↓ fructokinase gene	↑ melatonin-induced sugar accumulation ↑ growth inhibition	[81]
*Nicotiana tabacum* (in vitro)	0.2	↑ starch ↑ PEPCK and α-amylase genes	↑ sugar starved ↑ gluconeogenesis	[82]
Chinese hickory (plants)	100	↑ total soluble sugars, starch ↑ proline	↑ drought tolerance ↑ photosynthesis, transpiration	[83]
*Arabidopsis thaliana* (Pseudomonas syringe infected)	20	↑ fructose, glucose, melibose, sucrose, maltose, galatose, tagatofuranose and glycerol	↑ bacterial innate immunity ↑ disease resistance	[73]

↑, Increased content or increased action; ↓, Decreased content or decreased action.

## Data Availability

Not applicable.

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
