# Peer review of "Melatonin and Carbohydrate Metabolism in Plant Cells"

_plants, 2021, doi:10.3390/plants10091917_

Round 1

Reviewer 1 Report

This review provides a targeted overview of melatonin interactions with carbohydrate metabolism and is prepared by experts in the field making it worthy of consideration, however, some revisions are required to meet the authors stated goals of the review and address technical errors (including a typo check and proof read) prior to publication.

Abstract

- The link between stress and carbohydrates is not clear, revision of the second and third sentences of the abstract would help to clarify this so they do not appear disjunct.

Introduction

  • Line 34 typo correct to synthesized
  • Line 38-41 last sentence of intro. Could a description of the criteria for literature inclusion in the review be included?

Section 2 Biosynthesis

  • This section addresses only the primary melatonin biosynthetic pathway in plants. Please clarify this is the case. Additionally, significant new research has established that several alternate pathways are biologically relevant in plants for melatonin biosynthesis. These should be mentioned.
  • Please add references to support statements in lines 49-56.
  • Line 57 correct to "involved"
  • Line 58 please add relevant reference
  • Line 59, for which enzyme(s) are the mitochondria implicated?

Roles of melatonin in plants

  • This section relies heavily on citation of previous reviews by the authors vs original research. A greater emphasis on original research would help to strengthen the article
  • Line 70-71 please include reference
  • Line 72-73 reference

Table 1 / Line 101 - are the articles included comprehensive or a subset of the literature? If a subset please include criteria used for selection.

Section 4 Effect of melatonin in simple carbohydrates...

  • Can effects on photosynthesis and senescence be separated out from carbohydrates while the two are inherently linked but it is not clear in all instances if the effects of melatonin are due to improving photosynthetic capacity due to its antioxidant status (as discussed later in the article) or via more carbohydrate-specific mechanisms?
  • Line 110 "As well as to an improvement" remove "to"
  • Line 111 "Othef" -> "other
  • Line 123, please expand on the other fruits, maybe this could be a subsection?

Section 5 - Regulatory action...

  • Line 142 "waterloging" -> "waterlogging"

Section 6 - Conclusions

  • Line 189 correct to "provided"
  • Line 190 Is conical the intended word?
  • Please add references to support statements on postharvest effects as this is the section where this is primarily discussed. Especially lines 211-213 and 216-217

Author Response

see file

Reviewer 2 Report

The review discusses the role of melatonin in regulating primary carbohydrate metabolism in plant cells. This review extends the melatonin’s current role as plant stress modulator. It describes a number of studies conducted in past few years show modulatory role of melatonin in biosynthesis and degradation of plant carbohydrates including monosaccharides, disaccharides, polysaccharides and their derivatives. A number of the studies on carbohydrates and melatonin described in the review are carried out in crop plants, implicating the practical importance of this research. The regulatory action of melatonin in carbohydrate metabolism is a new area of research, there are still a lot of questions that need to be answered before it can be established as modulator of carbohydrate metabolism in plants.  

The review is well written, easy to follow and well-organized. Here are few suggestions for the authors:

  1. In figure 3, the organelles are not labelled, so please label the organelles. Also in figure 3 legend, line 186, correct spelling of “of”, currently it is written as “o”.
  2. There are few minor spelling errors e.g., in line 111, Other is spelled as Othef and in line 189, provided is spelled as providied, so please conduct a spell check on the text.

Author Response

see file
